# ACCELERATED VALUE ITERATION VIA ANDERSON MIXING

## ABSTRACT

Acceleration for reinforcement learning methods is an important and challenging theme. We introduce the Anderson acceleration technique into the value iteration, developing an accelerated value iteration algorithm that we call Anderson Accelerated Value Iteration (A2VI). We further apply our method to the Deep Q-learning algorithm, resulting in the Deep Anderson Accelerated Q-learning (DA2Q) algorithm. Our approach can be viewed as an approximation of the policy evaluation by interpolating on historical data. A2VI is more efficient than the modified policy iteration, which is a classical approximate method for policy evaluation. We give theoretical analysis of our algorithm and conduct experiments on both toy problems and Atari games. Both the theoretical and empirical results show the effectiveness of our algorithm.

## 1    INTRODUCTION

In reinforcement learning (Sutton & Barto, 1998), an agent seeks for the optimal policy in a specific sequential decision problem. Several algorithms have been proposed over the course of time, including the famous Q-learning (Watkins & Dayan, 1992), SARSA (Rummery & Niranjan, 1994; Sutton & Barto, 1998), and policy gradient methods (Sutton et al., 2000). In complicated decision problems where tabular representations are intractable, function approximations are usually used for estimating state-action values (Kaelbling et al., 1996; Sutton & Barto, 1998; Sutton et al., 2000). Inspired by the success of deep learning, Deep Q-Learning (DQN) (Mnih et al., 2013) and its variants (Bellemare et al., 2017; Schaul et al., 2015; Van Hasselt et al., 2016; Wang et al., 2015) utilize a deep neural network as the value approximator, which has successfully solved end-to-end decision problems such as Atari2000.

The value iteration (VI) Bellman (1957) and policy iteration (PI) (Howard, 1964) are the most classical methods for value updating. The main difference between them is that PI evaluate the current policy accurately during the iteration while VI does not. Thanks to the accurate evaluation of the current policy, policy iteration uses significantly less policy improvement steps to converge to the optimal value. Although PI has a faster convergence rate than VI, most of the existing methods employ a rather slow value iteration procedure, because thoroughly evaluating a policy is costly or even intractable under complex environments. To retain the fast convergence property of policy iteration while reducing its computation overhead, researchers have proposed several modifications to the original policy iteration (Alla et al., 2015; Puterman, 1994). The modified policy iteration (MPI) method (Puterman, 1994) tries to deal with this problem by approximating the solution to policy evaluation via the truncated Neumann expansion of an inverse matrix. However, this approximation requires extra iterative steps, which is still computationally inefficient for complex decision problems where sampling is costly, compared with the value iteration procedure where the policy iteration step is skipped.

Interpolation methods have been widely used in first order optimization problems (Bubeck et al., 2015; Scieur et al., 2016; 2017; Xie et al., 2018). These methods extract information from historical data and are proven to converge faster than vanilla gradient methods. However, the interpolation method is not widely applied in

reinforcement learning. The most recent work related to interpolation is the averaged-DQN (Anschel et al., 2016), which calculates the average Q-value over the history and demonstrated that such an operation is effective for variance reduction.

Acceleration in value iteration and policy iteration has attracted researchers' great attention. Classical methods for accelerating value iteration include Gauss-Seidel value iteration (Puterman, 1994) and Jacobi value iteration (JAC) (Puterman, 1994). More recently, Alla et al. (2015) proposed an acceleration method that switches between a coarse-mesh value iteration and a fine-mesh policy iteration during different stages. Laurini et al. (2016) performed a Jacobi-like acceleration method on dynamic programming problems. In a recent work (Laurini et al., 2017), the value iteration procedure is accelerated by only updating a part of the values. None of the previous methods have proposed acceleration methods with an application of interpolation.

In this paper, to solve the policy evaluation problem more efficiently, we propose an alternative algorithm based on multi-step interpolation. Explicitly, the solution to the policy evaluation problem is approximately represented by a weighted combination of historical values, whose weights are adaptively updated by an optimization procedure. To reduce the computational complexity, we resort to the Anderson mixing method (Anderson, 1965; Walker & Ni, 2011; Toth & Kelley, 2015) to do the approximation with only a short length of history. Our approach fits the gap between value iteration and policy iteration, ending in an updating rule without adding much extra computational complexity to the original value iteration procedure. We also extend this approach to deep reinforcement learning problems.

The remainder of this paper is organized as follows. In Section 2, we introduce the foundations of reinforcement learning and present typical value updating algorithms. In Section 3, we derive the Anderson accelerated methods. In Section 4, we give a theoretical analysis of the convergence of our method. In Section 5, we test our method in different environments and empirically show the effectiveness of it. Finally, we conclude our work in Section 6.

## 2 PRELIMINARIES

In this paper we mainly consider a finite-state and finite-action scenario in reinforcement learning. In this case, an Markov Decision Process (MDP) system is defined by a 5-tuple $(\mathcal{S}, \mathcal{A}, P, \boldsymbol{r}, \gamma)$, where $\mathcal{S}$ is a finite state space, $\mathcal{A}$ is a finite action space, $P \in \mathbb{R}^{(|\mathcal{S}| \times |\mathcal{A}|) \times |\mathcal{S}|}$ is the collection of state-to-state transition probabilities, $\boldsymbol{r} \in \mathbb{R}^{|\mathcal{S}| \times |\mathcal{A}|}$ is the reward matrix, $\gamma$ is the discount factor. A policy $\pi \in \mathcal{A}^{|\mathcal{S}|}$ is a vector of actions at each state. The transition matrix $P_\pi \in \mathbb{R}^{|\mathcal{S}| \times |\mathcal{S}|}$ and reward vector $\boldsymbol{r}_\pi \in \mathbb{R}^{|\mathcal{S}|}$ under policy $\pi$ are defined as $P_\pi(i, j) = P((i, \pi(i)), j), \boldsymbol{r}_\pi(i) = \boldsymbol{r}(i, \pi(i))$. We further define the value $\boldsymbol{v}^\pi \in \mathbb{R}^{|\mathcal{S}|}$ and the Q-value $\boldsymbol{q}^\pi \in \mathbb{R}^{|\mathcal{S}| \times |\mathcal{A}|}$ under a given MDP and policy, where each element of $\boldsymbol{v}^\pi$ and $\boldsymbol{q}^\pi$ is defined as

$$\boldsymbol{v}^\pi(s) = \mathbb{E}_{s_0 = s, s_{t+1} \sim P_\pi(s_t, \ldots)} \sum_{t=0}^{\infty} \gamma^t \boldsymbol{r}_\pi(s_t),$$

$$\boldsymbol{q}^\pi(s, a) = \boldsymbol{r}(s, a) + \mathbb{E}_{s_1 \sim P((s,a),\ldots), s_{t+1} \sim P_\pi(s_t, \ldots)} \sum_{t=1}^{\infty} \gamma^t \boldsymbol{r}_\pi(s_t).$$

We can verify that $\boldsymbol{q}^\pi = \boldsymbol{r} + \gamma P \boldsymbol{v}^\pi$. We define $\boldsymbol{q}_{\tilde{\pi}}^\pi \in \mathbb{R}^{|\mathcal{S}|}$ by $\boldsymbol{q}_{\tilde{\pi}}^\pi(i) = \boldsymbol{q}^\pi(i, \tilde{\pi}(i))$, and say a vector to be the maximum among a set if each entry of it is bigger than that of the other vectors. The values satisfy the Bellman equation:

$$\boldsymbol{v}^\pi = \Gamma_\pi(\boldsymbol{v}^\pi) = \boldsymbol{r}_\pi + \gamma P_\pi \boldsymbol{v}^\pi.$$

The policy $\pi^* = \operatorname{argmax}_\pi \boldsymbol{q}^\pi$ is called the optimal policy, whose value or Q-value is denoted as $\boldsymbol{v}^*$ or $\boldsymbol{q}^*$. Note that $\boldsymbol{v}^*$ satisfies the Bellman optimality equation

$$\boldsymbol{v}^* = \Gamma(\boldsymbol{v}^*) = \max_\pi(\boldsymbol{r}_\pi + \gamma P_\pi \boldsymbol{v}^*).$$

Therefore, finding the optimal policy is equivalent to finding the fixed point of the operator $\Gamma(\boldsymbol{v})$.

## 2.1 FIXED POINT ITERATION METHODS

Value iteration (VI) is the most widely used and best-understood algorithm for solving Markov decision problems. It solves the fixed point problem by iterating the following steps repeatedly,

$$\boldsymbol{v}^{(t+1)} = \Gamma(\boldsymbol{v}^{(t)}) = \max_\pi(\boldsymbol{r}_\pi + \gamma P_\pi \boldsymbol{v}^{(t)}).$$

An alternative solution is policy iteration (PI), which maintains both the value $\boldsymbol{v}^{(t)}$ and the policy $\pi^{(t)}$ during each iteration. The procedure alternatively iterates the following two steps:

- Policy evaluation: Find a $\boldsymbol{v}^{(t)}$ such that

$$\boldsymbol{v}^{(t)} = \Gamma_{\pi^{(t)}}(\boldsymbol{v}^{(t)}) = \boldsymbol{r}_{\pi^{(t)}} + \gamma P_{\pi^{(t)}} \boldsymbol{v}^{(t)}, \tag{1}$$

  which can be directly computed by

$$\boldsymbol{v}^{(t)} = (I - \gamma P_{\pi^{(t)}})^{-1} \boldsymbol{r}_{\pi^{(t)}}. \tag{2}$$

- Policy improvement: Improve the current policy by

$$\pi^{(t+1)} = \operatorname{argmax}_\pi(\boldsymbol{r}_\pi + \gamma P_\pi \boldsymbol{v}^{(t)}).$$

Theoretical analysis has shown that VI enjoys a $\gamma$-linear convergence rate (i.e., $\|\boldsymbol{v}^{(t)} - \boldsymbol{v}^*\|_\infty \leq \gamma \|\boldsymbol{v}^{(t-1)} - \boldsymbol{v}^*\|_\infty$), while PI converges much faster with $\|\boldsymbol{v}^{(t)} - \boldsymbol{v}^*\|_\infty \leq K \|\boldsymbol{v}^{(t-1)} - \boldsymbol{v}^*\|_\infty^2$ (Puterman, 1994), where $K$ is some constant related with $\gamma$ and the given MDP. Both VI and PI are model-based, because the greedy policy cannot be determined when $\boldsymbol{r}$ and $P$ are unknown. The VI under $\boldsymbol{q}$-notation is well-known as Q-learning (Watkins & Dayan, 1992). We will analyze our method under $\boldsymbol{v}$-notation, but our analysis also works under the corresponding $\boldsymbol{q}$-notation.

The main difference between VI and PI is whether the current policy is fully evaluated. Though PI converges faster than VI, this advantage diminishes under several settings. In reinforcement learning, we can only access an oracle that returns the reward and next state given the current state and selected action. Under such a setting, each value iteration step can be performed by estimating $\Gamma(\boldsymbol{v})$ through sampling. But the policy evaluation step based on equation (2) becomes intractable because it is quite time-consuming to compute $(I - \gamma P_{\pi^{(t)}})^{-1}$. The modified policy iteration method (Puterman & Brumelle, 1978) partially solves the problem by setting $\boldsymbol{v}^t \approx (\Gamma_{\pi^{(t)}})^{m_t}(\boldsymbol{v}^{(t-1)})$ where $m_t$ is a (possibly large) integer related to $t$. However, this method requires to evaluate a series of values $(\Gamma_{\pi^{(t)}})^i(\boldsymbol{v}^{(t-1)})$ for $i = 1, 2, \ldots, m_t$, which is computationally inefficient.

## 3 ANDERSON ACCELERATED VALUE ITERATION

Based on the observation that full policy evaluation accelerates convergence, we propose an approximate policy evaluation method. The method aims to approximately solve the policy evaluation problem, circumventing the matrix inversion and iterative procedures mentioned above.

We first utilize the linearity of equation (1), defining $B_\pi(v) = \Gamma_\pi(v) - v$ and converting the problem into an equivalent form of solving the equation $B_\pi(v) = \mathbf{0}$. Suppose we have obtained a set of values $B_\pi(v^1), B_\pi(v^2), \ldots, B_\pi(v^k)$ with respect to $v^1, v^2, \ldots, v^k$. Consider to find a set of weights $\boldsymbol{\alpha} = (\alpha_1, \alpha_2, \ldots, \alpha_k)^T$, subject to $\sum_{i=1}^{k} \alpha_i = 1$, which satisfies that

$$\sum_{i=1}^{k} \alpha_i B_\pi(v^i) = \mathbf{0}.$$

Then the combination $\tilde{v} = \sum_{i=1}^{k} \alpha_i v^i$ will satisfy the following relationship:

$$B_\pi(\tilde{v}) = r_\pi + \gamma P_\pi \tilde{v} - \tilde{v} = \sum_{i=1}^{k} \alpha_i(r_\pi + \gamma P_\pi v^i - v^i) \tag{3}$$

$$= \sum_{i=1}^{k} \alpha_i B_\pi(v^i) = \mathbf{0}. \tag{4}$$

This relation implies $\tilde{v}$ can be viewed as an approximate solution to equation (1) provided the sampling estimations are accurate enough. However, this step needs to keep track of the previous values and recompute $\Gamma_\pi$ on all of them. To reduce the huge memory usage and computation, we choose $v^i$ from the recent history, i.e., $v^i = v^{(t-i)}, i = 1, 2, \ldots, k$, and replace $B_{\pi^{(t)}}(v^{(t-i)})$ with the previously computed values $B_{\pi^{(t-i)}}(v^{(t-i)})$. This modification is based on the observation that the recent successive policies do not change sharply and therefore $B_{\pi^{(t-i)}}(v^{(t-i)}) \approx B_{\pi^{(t)}}(v^{(t-i)})$. This modification approximately solves the policy evaluation problem without model estimation or extra function evaluations.

Another critical issue is that we cannot guarantee the existence of $\boldsymbol{\alpha}$ given that $k$ is small, because the dimension of $B_\pi(v)$ is usually much higher than $k$. Inspired by the Anderson acceleration technique (Anderson, 1965; Ortega & Rheinboldt, 1970; Walker & Ni, 2011), we instead look for a combination of $\{B_{\pi^{(t-i)}}(v^{(t-i)})\}_{i=1}^{k}$,

$$\boldsymbol{\alpha}^{(t)} = \operatorname*{argmin}_{\boldsymbol{\alpha} \in \Omega \cap \Lambda} \|B^{(t)} \boldsymbol{\alpha}\|, \tag{5}$$

where $B^{(t)} = (B_{\pi^{(t-1)}}(v^{(t-1)}), B_{\pi^{(t-2)}}(v^{(t-2)}), \ldots, B_{\pi^{(t-k)}}(v^{(t-k)}))$, $\Omega = \{\boldsymbol{\alpha} | \mathbf{1}^T \boldsymbol{\alpha} = 1\}$, $\Lambda$ is an extra constraint on the values attainable by $\boldsymbol{\alpha}$. Typically, $\Lambda$ can be chosen from the following forms:

- Total space, $\Lambda_{\text{tot}} = \mathbb{R}^k$;
- Boxing constraint, $\Lambda_{\text{box}} = \{\boldsymbol{\alpha} | -m\mathbf{1} \leq \boldsymbol{\alpha} \leq m\mathbf{1}\}$;
- Convex combination constraint, $\Lambda_{\text{cvx}} = \{\boldsymbol{\alpha} | \mathbf{0} \leq \boldsymbol{\alpha} \leq \mathbf{1}\}$;
- Extrapolation constraint, $\Lambda_{\text{exp}} = \{\boldsymbol{\alpha} | \alpha_1 \geq 1, \alpha_i \leq 0, i = 2, 3, \ldots, k\}$.

When the $\ell_2$ norm is used and $\Lambda = \Lambda_{\text{tot}}$, the solution can be written explicitly as $\boldsymbol{\alpha}^{(t)} = [(B^{(t)})^\top B^{(t)}]^{-1} \mathbf{1} / \mathbf{1}^\top [(B^{(t)})^\top B^{(t)}]^{-1} \mathbf{1}$, whose derivation is placed in the appendix. Note that if we simply set $v^{(t)} = \sum_{i=1}^{k} \alpha_i^{(t)} v^{(t-i)}$, the values will always locate in the subspace expanded by historical values $v^{(t-1)}, v^{(t-2)}, \ldots, v^{(t-k)}$. When the solution to equation (1) does not lie in such a subspace, there is no hope for convergence with application of such updating rule directly. To jump out of the subspace, we perform an extra value iteration step to this combination. Then we will get the updated value,

$$v^{(t)} = \max_{\pi} \left( r_\pi + \gamma P_\pi \left[ \sum_{i=1}^{k} \alpha_i^{(t)} v^{(t-i)} \right] \right).$$

## 3.1 The Algorithm

Based on our previous discussion, we present the $k$-step Anderson Accelerated Value Iteration (A2VI) in Algorithm 1. In the first $k$ steps, the value is updated according to the original VI. Otherwise, we perform an interpolation procedure, where the weights are attained from solving the problem (5). The original value iteration algorithm can be viewed as a special case of our algorithm with $k = 1$.

---

**Algorithm 1** Anderson Accelerated Value Iteration (A2VI)

---

**Input:** $\boldsymbol{v}^{(0)}, P, \boldsymbol{r}, \gamma, k, T$
1: **for** $t = 1, 2, \ldots, T$ **do**
2:     $B_{\pi^{(t-1)}}(\boldsymbol{v}^{(t-1)}) = \max_\pi (\boldsymbol{r}_\pi + \gamma P_\pi \boldsymbol{v}^{(t-1)}) - \boldsymbol{v}^{(t-1)}$
3:     **if** $t < k$ **then**
4:         $\boldsymbol{v}^{(t)} = \max_\pi (\boldsymbol{r}_\pi + \gamma P_\pi \boldsymbol{v}^{(t-1)})$
5:     **else**
6:         Calculate $(\alpha_1^{(t)}, \alpha_2^{(t)}, \ldots, \alpha_k^{(t)})$ by solving the optimization problem (5)
7:         $\boldsymbol{v}^{(t)} = \max_\pi \left( \boldsymbol{r}_\pi + \gamma P_\pi \left[ \sum_{i=1}^k \alpha_i^{(t)} \boldsymbol{v}^{(t-i)} \right] \right)$
8:     **end if**
9: **end for**
10: $\pi^{(T)} = \operatorname{argmax}_\pi (\boldsymbol{r}_\pi + \gamma P_\pi \boldsymbol{v}^{(T)})$
11: **return** $\boldsymbol{v}^{(T)}, \pi^{(T)}$

---

Both Anderson Acceleration (AA) and A2VI have the same spirit of interpolating on historical data. However, A2VI does not straightforwardly apply AA to the Bellman optimality equation. Note that AA has the updating rule $\boldsymbol{v}^t = \sum \alpha_i B(\boldsymbol{v}^{t-i})$, while A2VI exchange the order of the operator $\texttt{sum}$ and $B(\cdot)$ due to the motivation from equation (3). This exchange puts the nonsmooth operator $\texttt{max}$ out of the affine combination, simplifying the theoretical analysis.

We present a geometric explanation on the iterative steps of VI, PI, A2VI under 1-dimensional case in Figure 1. In value iteration, $\boldsymbol{v}^{(t)}$ is attained by making a vertical line at $(\boldsymbol{v}^{(t-1)}, \boldsymbol{0})$, finding its intersection with the function line at $(\boldsymbol{v}^{(t-1)}, B(\boldsymbol{v}^{(t-1)}))$, then drawing a line with slope $-1$ through $(\boldsymbol{v}^{(t-1)}, B(\boldsymbol{v}^{(t-1)}))$ and finding its intersection with the horizon axis at $(\boldsymbol{v}^{(t)}, \boldsymbol{0})$; In policy iteration, $\boldsymbol{v}^{(t)}$ is attained by first getting $(\boldsymbol{v}^{(t-1)}, B(\boldsymbol{v}^{(t-1)}))$ in the same way as value iteration, then calculating the tangent line through $(\boldsymbol{v}^{(t-1)}, B(\boldsymbol{v}^{(t-1)}))$ and finding its intersection with the horizon axis. In Anderson accelerated value iteration, each step is first performed in a similar style to policy iteration except that the tangent line is replaced with a secant line. Then a value iteration step is performed to get $\boldsymbol{v}^{(t)}$.

From the figure, we can see that VI only utilizes the current value of the Bellman residual, while PI is similar to Newton's method Puterman & Brumelle (1978), utilizing the gradient information to achieve a faster convergence rate. Our method serves as an intermediate between them, each step of which is composed of an ordinary value iteration step and a secant step. In specific, the replacement of the tangent line to a secant line can be viewed as a quasi-Newton's method, which is shown computationally more efficient while keeping a fast convergence rate in several particular settings. Both PI and A2VI converge to the fixed point in a smaller number of steps than VI. Compared with PI, A2VI is more practical because it approximates the tangent line by a secant line, which circumvents the costly model estimation step.

### 3.2 Extension to Model-free Learning Algorithm

We can rewrite our algorithm under $\boldsymbol{q}$-notation, and get the Anderson Accelerated Q-Learning (A2Q) Algorithm shown in the appendix. Combined with the technique of deep learning, our method can be applied to end-to-end decision problems, resulting in the Deep Anderson Accelerated Q-Learning (DA2Q) Algorithm (Algorithm 2).

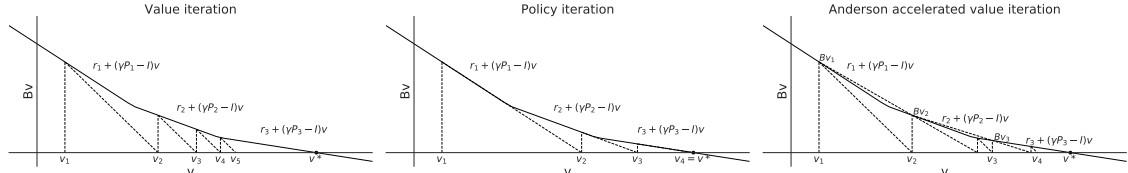

Figure 1: Geometric interpolation of VI, PI and A2VI.

---

**Algorithm 2** Deep Anderson Accelerated Q-learning (DA2Q)

---

**Input:** $M, N, T, \gamma, b, B, \varepsilon, \eta, K, C$
1: Initialize replay memory $\mathcal{D}$ to capacity $N$, initialize Q-value function $Q$ with random weights $\theta$
2: $\theta_{-k} = \theta, \alpha_1 = 1, \alpha_k = 0$ for $k = 2, ..., K$
3: $s = 0$
4: **for** episode = $1, 2, \ldots, M$ **do**
5:      Initialize $s_1 \sim \rho(s)$
6:      **for** $t = 1, 2, \ldots, T$ **do**
7:          With probability $\varepsilon$ select a random action $a_t$, otherwise select $a_t = \arg\max_a Q(s_t, a; \theta)$
8:          Execute action $a_t$, observe reward $r_t$ and state $s_{t+1}$
9:          Store transition $(s_t, a_t, r_t, s_{t+1})$ in $\mathcal{D}$
10:          Sample a random minibatch of transitions $\{(s_j, a_j, r_j, s'_j)\}_{j=1}^b$ from $\mathcal{D}$
11:          **for** $j = 1, 2, \ldots, b$ **do**
12: 
$$y_j = \begin{cases} r_j & \text{for terminal state} \\ r_j + \gamma \max_a \left( \sum_{k=1}^K \alpha_k Q(s'_j, a; \theta_{-k}) \right) & \text{for non-terminal state} \end{cases}$$
13:          **end for**
14:          $L(\theta) = \frac{1}{b} \sum_{j=1}^b (y_j - Q(s_j, a_j; \theta))^2$
15:          $\theta = \theta - \eta \frac{\partial L}{\partial \theta}$
16:          $s = s + 1$
17:          **if** $s \bmod C = 0$ **then**
18:             Assign $\theta_{-k} = \theta_{-(k-1)}$ for $k = K, K-1, \ldots, 2$. Assign $\theta_{-1} = \theta$.
19:             **if** $s \geq K(C-1)$ **then**
20:                 Sample a random minibatch of transitions $\{(s_j, a_j, r_j, s'_j)\}_{j=1}^B$ from $\mathcal{D}$
21:                 **for** $j = 1, 2, \ldots, B$ **do**
22:                      **for** $k = 1, 2, \ldots, K$ **do**
23: 
$$d_j^k = \begin{cases} r_j - Q(s_j, a_j; \theta_{-k}) & \text{for terminal state} \\ r_j + \gamma \max_a Q(s'_j, a; \theta_{-k}) - Q(s_j, a_j; \theta_{-k}) & \text{for non-terminal state} \end{cases}$$
24:                      **end for**
25:                 **end for**
26:             **end if**
27:             $(\alpha_1, \alpha_2, \ldots, \alpha_K) = \text{argmin}_{(\alpha_1, \alpha_2, \ldots, \alpha_K)} \sum_{j=1}^B (\sum_{k=1}^K \alpha_k d_j^k)^2$ s.t. $\sum_{k=1}^K \alpha_k = 1$
28:          **end if**
29:      **end for**
30: **end for**

---

## 4 THEORETICAL ANALYSIS

We first analyze of the local convergence of the A2VI algorithm under boxing constraint. Our result shows that in a small neighborhood of the optimal value, our algorithm enjoys an exponential convergence rate.

**Theorem 1.** *For any MDP with a unique optimal policy, there exists some $\delta > 0$, such that for any initial value $\boldsymbol{v}^{(0)} \in U_\delta(\boldsymbol{v}^*) = \{v|\|v - v^*\|_\infty \leq \delta\}$, the A2VI algorithm under boxing constraint maintains the following properties:*

   (i)  $\|\Gamma(\boldsymbol{v}^{(t)}) - \boldsymbol{v}^{(t)}\|_\infty \leq \gamma\|\Gamma(\boldsymbol{v}^{(t-1)}) - \boldsymbol{v}^{(t-1)}\|_\infty, \forall t = 1, 2, \ldots;$

   (ii)  $\|\boldsymbol{v}^{(t)} - \boldsymbol{v}^*\|_\infty \leq \frac{\gamma^t}{1-\gamma}\|\Gamma(\boldsymbol{v}^{(0)}) - \boldsymbol{v}^{(0)}\|_\infty, \forall t = 1, 2, \ldots.$

Generally, it is difficult to obtain the global convergence rate of A2VI, since the operation $\max$ is nonsmooth. To guarantee the convergence, we introduce a rejection step to the original algorithm. We say $\boldsymbol{v}$ is monotonic improving if $\Gamma(\boldsymbol{v}) \geq \boldsymbol{v}$, and denote the set of such values as $V_B$. We propose the A2VI algorithm with the rejection step, which only differs with Algorithm 1 at line 6. After calculating $\boldsymbol{\alpha}^{(t)}$, we test whether the affine combination $\sum_{i=1}^{k} \alpha_i^{(t)} \boldsymbol{v}^{t-i}$ lies in $V_B$. If the answer is negative, the interpolation step will be replaced with an ordinary value iteration step. We put the pseudocode of A2VI with the rejection step in Appendix. With this modification, we can have the following convergence properties.

**Theorem 2.** *For the A2VI algorithm with the rejection step with $\Lambda = \Lambda_{\text{cvx}}$, if $\boldsymbol{v}^{(0)} \in V_B$, then we have*
$$\boldsymbol{v}^{(t)} \in V_B, \|\Gamma(\boldsymbol{v}^{(t)}) - \boldsymbol{v}^{(t)}\| \leq \gamma\|\Gamma(\boldsymbol{v}^{(t-1)}) - \boldsymbol{v}^{(t-1)}\|, \forall t = 1, 2, \ldots$$

**Theorem 3.** *For the A2VI algorithm with $\Lambda = \Lambda_{\text{exp}}$, if $\boldsymbol{v}^0 \geq \boldsymbol{0}$ and $\boldsymbol{v}^{(0)} \in V_B$, then we have*

   (a)  *Monotone improving values, $\boldsymbol{v}^{(t-1)} \leq \boldsymbol{v}^{(t)} \leq \boldsymbol{v}^*, \boldsymbol{v}^{(t)} \in V_B, \forall t = 1, 2, \ldots$*

   (b)  *$\gamma$-linear convergence rate, $\|\boldsymbol{v}^* - \boldsymbol{v}^{(t)}\|_\infty \leq \gamma\|\boldsymbol{v}^* - \boldsymbol{v}^{(t-1)}\|_\infty.$*

## 5   EXPERIMENTS

To validate the effectiveness of our method, we conduct several experiments.

### 5.1   EXPERIMENTS ON TOY MODELS

We first test our method on three toy models. The first model is a randomly generated MDP with $|\mathcal{S}| = 100$ and $|\mathcal{A}| = 50$. The transition probabilities of the MDP are generated from a uniform distribution on $[0, 1]$, and the rewards are generated from a standard normal distribution. The second model is the $N$-Chain problem with $N = 100$, where a reward of $0.1$ is given at state $0$ and a reward of $1$ is given at state $N$. At each state, the agent can either choose to move forward or backward, and will move to the selected direction with probability $0.9$ and to the opposite direction with probability $0.1$. The last model is a $20 \times 20$ Gridworld model, where a reward of $1$ is given at state $(20, 20)$. At each state, the agent can choose one of the $4$ directions and will move to that direction with probability $0.7$, or move to one of the other directions with probability $0.1$ for each. We perform the standard value iteration, policy iteration and Anderson accelerated value iteration with/without the rejection step on these models. In our experiment, each policy iteration step is approximately solved by the modified policy iteration method with $100$ inner iterations. To compare our method with the averaged updating scheme (Anschel et al., 2016), we further construct and compare our algorithm with the averaged value iteration. The value of $\|\boldsymbol{v}^t - \boldsymbol{v}^*\|$ w.r.t. step $t$ is shown in Figure 2, where the results are averaged from $30$ independent experiments.

From the results we can see that the policy iteration converges fastest for all of the three models, however, since each of its steps includes $100$ inner iterations, the actual computation cost is very high. Among value iteration methods, the Anderson accelerated value iteration converges fastest. The acceleration effect is remarkable in randomly generated MDPs, but A2VI slows down at the first few steps in the latter two experiments. However, adding a rejection step solves the problem and attains a faster convergence rate. Another observation is that in the toy model case, the averaged value iteration cannot be used for acceleration.

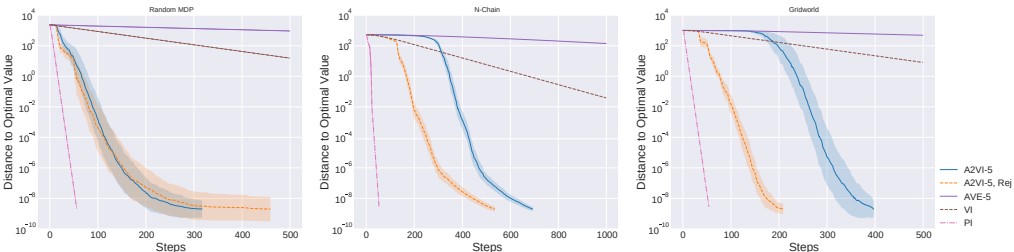

Figure 2: Experiment results on several toy models.

## 5.2 Experiments on Atari Games with Deep Learning Based Techniques

To figure out the performance of our method on complex environments, we apply our method to Atari games from Gym (Brockman et al., 2016), which is a Python API to Arcade Learning Environment (Bellemare et al., 2013). We compare DA2Q with DQN (Mnih et al., 2013) and Averaged-DQN (Anschel et al., 2016). Details of the experiment settings are given in Appendix D.

As Figure 3 points out, our algorithm DA2Q obtains a significant improvement over both the original DQN algorithm and the Averaged DQN algorithm. When compared with other interpolation method such as Averaged-DQN, the overall performance of our method also tends to be stabler, always being superior than other methods among all of the three environments, while the performance of Averaged-DQN varies a lot.

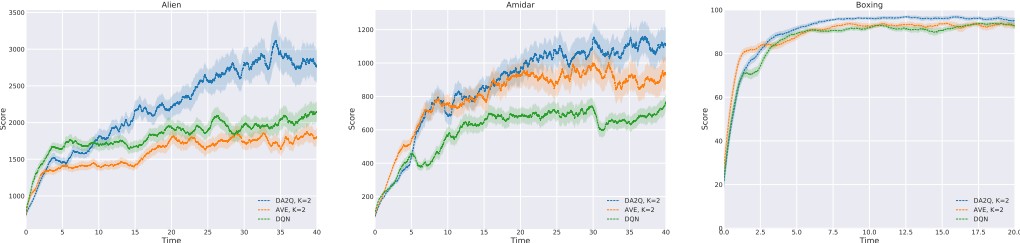

Figure 3: Training Performance on Atari games, score is smoothed with 250 windows while the shaded area is the 0.25 standard deviation.

Compared with DQN, the extra computational cost is actually low, since the $\alpha$ is updated only once every $C$ steps, which only involves an inversion on a very small-size matrix ($k \times k$). The $k$ target values are computed parallelly in the TensorFlow (Abadi et al., 2016), which cost the same time as in DQN. Moreover, the extra runtime can be ignored when compared with the costly back propagations and interaction with environments.

## 6 Conclusion

We have proposed the Anderson accelerated value iteration method, which is a novel acceleration approach for reinforcement learning. We have proved the convergence property of our method under certain conditions. Our algorithm empirically achieves a superior performance on toy models and several Atari games. Despite the success of our algorithm, several questions remain open. The convergence analysis for the general case is lacking, and we only provide convergence guarantees but do not give a theoretical analysis of the acceleration effect of A2VI, which we leave for future work.

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

## APPENDIX

### A. A2Q

Here is the pseudocode of A2Q, which is A2VI in $\boldsymbol{q}$-notation.

---
**Algorithm 3** Anderson Accelerated Q-Learning (A2Q)
---
**Input:** $\boldsymbol{q}^0, P, \boldsymbol{r}, \gamma, k, T$
1: **for** $t = 1, 2, \ldots, T$ **do**
2:      $B_{\pi^{(t-1)}}(\boldsymbol{q}^{t-1}) = \mathrm{vec}(\boldsymbol{r}) + \gamma P \max_\pi \boldsymbol{q}_\pi^{t-1} - \mathrm{vec}(\boldsymbol{q}^{t-1})$
3:      **if** $t < k$ **then**
4:          $\mathrm{vec}(\boldsymbol{q}^t) = \mathrm{vec}(\boldsymbol{r}) + \gamma P \max_\pi \boldsymbol{q}_\pi^{t-1}$
5:      **else**
6:          Calculate $(\alpha_1, \alpha_2, \ldots, \alpha_k)$ by solving the optimization problem (5)
7:          $\mathrm{vec}(\boldsymbol{q}^t) = \mathrm{vec}(\boldsymbol{r}) + \gamma P \max_\pi (\sum_{i=1}^{k} \alpha_i \boldsymbol{q}_\pi^{t-i})$
8:      **end if**
9: **end for**
10: $\pi^T = \arg\max_\pi \boldsymbol{q}_\pi^T$
11: **return** $\boldsymbol{q}^T, \pi^T$

---

### B. A2VI WITH THE REJECTION STEP

Here is the pseudocode of A2VI with the rejection step, the only difference between this algorithm with algorithm 1 is before the interpolation step, we first check whether the affine combination is in $V_B$. If the answer is negative, then this interpolation step is replaced with an ordinary value iteration step.

---
**Algorithm 4** Anderson Accelerated Value Iteration with the Rejection Step
---
**Input:** $\boldsymbol{v}^{(0)}, P, \boldsymbol{r}, \gamma, k, T$
1: **for** $t = 1, 2, \cdots, T$ **do**
2:      $B_{\pi^{(t-1)}}(\boldsymbol{v}^{(t-1)}) = \max_\pi (\boldsymbol{r}_\pi + \gamma P_\pi \boldsymbol{v}^{(t-1)}) - \boldsymbol{v}^{(t-1)}$
3:      **if** $t < k$ **then**
4:          $\boldsymbol{v}^{(t)} = \max_\pi (\boldsymbol{r}_\pi + \gamma P_\pi \boldsymbol{v}^{(t-1)})$
5:      **else**
6:          Calculate $(\alpha_1^{(t)}, \alpha_2^{(t)}, \cdots, \alpha_k^{(t)})$ by solving the optimization problem (5)
7:          $\tilde{\boldsymbol{v}} = \sum_{i=1}^{k} \alpha_i^{(t)} \boldsymbol{v}^{(t-i)}$
8:          **if** $\max_\pi (\boldsymbol{r}_\pi + \gamma P_\pi \tilde{\boldsymbol{v}}) \geq \tilde{\boldsymbol{v}}$ **then**
9:              $\boldsymbol{v}^{(t)} = \max_\pi (\boldsymbol{r}_\pi + \gamma P_\pi \tilde{\boldsymbol{v}})$
10:          **else**
11:              $\alpha_1^{(t)} = 1, \alpha_i^{(t)} = 0$ for $i \neq 1$
12:              $\boldsymbol{v}^{(t)} = \max_\pi (\boldsymbol{r}_\pi + \gamma P_\pi \boldsymbol{v}^{(t-1)})$
13:          **end if**
14:      **end if**
15: **end for**
16: $\pi^{(T)} = \mathrm{argmax}_\pi (\boldsymbol{r}_\pi + \gamma P_\pi \boldsymbol{v}^{(T)})$
17: **return** $\boldsymbol{v}^{(T)}, \pi^{(T)}$

---

## C. PROOFS

**Lemma 1.** *For any MDP whose optimal policy is unique, there exists a $\delta > 0$ and a policy $\pi^*$ such that for any $\boldsymbol{v} \in U_\delta(\boldsymbol{v}^*) = \{\boldsymbol{v} | \|\boldsymbol{v} - \boldsymbol{v}^*\|_\infty \leq \delta\}$,*

$$\Gamma(\boldsymbol{v}) = \boldsymbol{r}_{\pi^*} + \gamma P_{\pi^*} \boldsymbol{v}$$

**Proof** Because the optimal policy is unique, for any nonoptimal policy $\pi$, for any state $s$ such that $\pi(s) \neq \pi^*(s)$ we have that $[\Gamma_{\pi^*}(\boldsymbol{v}^*)]_s > [\Gamma_\pi(\boldsymbol{v}^*)]_s$, $[\cdot]_s$ means executing operations on state $s$. Denote $A(\pi) = \{s | \pi(s) \neq \pi^*(s)\}$.

Suppose the optimal policy is $\pi^*$, then there exists $\varepsilon$ such that

$$\min_{\pi \neq \pi^*} \min_{s \in A(\pi)} [\Gamma(\boldsymbol{v}^*) - \Gamma_\pi(\boldsymbol{v}^*)]_s > \varepsilon > 0,$$

since the optimal policy is unique and the state space and the action space are finite. We choose $\delta = \frac{\varepsilon}{3\gamma}$, then for any policy $\pi$ and any $\boldsymbol{v} \in U_\delta(\boldsymbol{v}^*)$ we have

$$\|\Gamma_\pi(\boldsymbol{v}^*) - \Gamma_\pi(\boldsymbol{v})\|_\infty = \|\gamma P_\pi(\boldsymbol{v}^* - \boldsymbol{v})\|_\infty \leq \gamma \|\boldsymbol{v}^* - \boldsymbol{v}\|_\infty \leq \frac{\varepsilon}{3}$$

Then for any policy $\pi$, for any state $s \in A(\pi)$ , we have that

$$
\begin{aligned}
[\Gamma_{\pi^*}(\boldsymbol{v}) - \Gamma_\pi(\boldsymbol{v})]_s &= [(\Gamma_{\pi^*}(\boldsymbol{v}) - \Gamma_{\pi^*}(\boldsymbol{v}^*)) + (\Gamma_{\pi^*}(\boldsymbol{v}^*) - \Gamma_\pi(\boldsymbol{v}^*)) + (\Gamma_\pi(\boldsymbol{v}^*) - \Gamma_\pi(\boldsymbol{v}))]_s \\
&\geq \varepsilon - \|\Gamma_{\pi^*}(\boldsymbol{v}) - \Gamma_{\pi^*}(\boldsymbol{v}^*)\|_\infty - \|\Gamma_\pi(\boldsymbol{v}^*) - \Gamma_\pi(\boldsymbol{v})\|_\infty \\
&\geq \varepsilon - \frac{\varepsilon}{3} - \frac{\varepsilon}{3} \\
&= \frac{\varepsilon}{3}
\end{aligned}
$$

which means $\pi$ does not choose the optimal action in state $s$. Therefore, if $\pi$ selects the optimal action in every state $s \in \mathcal{S}$, then we must have $\pi_s = \pi_s^*, \forall s \in \mathcal{S}$, which implies $\pi^* \in \arg\max_\pi \Gamma_\pi(\boldsymbol{v})$, i.e., $\Gamma(\boldsymbol{v}) = \boldsymbol{r}_{\pi^*} + \gamma P_{\pi^*} \boldsymbol{v}$.

**Lemma 2.** *For any given MDP with optimal value $\boldsymbol{v}^*$ and any value $\boldsymbol{v}$, we always have*

$$(1 - \gamma)\|\boldsymbol{v} - \boldsymbol{v}^*\|_\infty \leq \|\Gamma(\boldsymbol{v}) - \boldsymbol{v}\|_\infty \leq (1 + \gamma)\|\boldsymbol{v} - \boldsymbol{v}^*\|_\infty.$$

**Proof**

$$
\begin{aligned}
\|\boldsymbol{v} - \boldsymbol{v}^*\|_\infty &= \|\boldsymbol{v} - \Gamma(\boldsymbol{v}) + \Gamma(\boldsymbol{v}) - \Gamma(\boldsymbol{v}^*)\|_\infty \\
&\leq \|\boldsymbol{v} - \Gamma(\boldsymbol{v})\|_\infty + \|\Gamma(\boldsymbol{v}) - \Gamma(\boldsymbol{v}^*)\|_\infty \\
&\leq \|\boldsymbol{v} - \Gamma(\boldsymbol{v})\|_\infty + \gamma \|\boldsymbol{v} - \boldsymbol{v}^*\|_\infty \\
\Rightarrow (1 - \gamma)\|_\infty \boldsymbol{v} - \boldsymbol{v}^*\|_\infty &\leq \|\Gamma(\boldsymbol{v}) - \boldsymbol{v}\|_\infty \\
\|\boldsymbol{v} - \boldsymbol{v}^*\|_\infty &= \|\boldsymbol{v} - \Gamma(\boldsymbol{v}) + \Gamma(\boldsymbol{v}) - \Gamma(\boldsymbol{v}^*)\|_\infty \\
&\geq \|\boldsymbol{v} - \Gamma(\boldsymbol{v})\|_\infty - \|\Gamma(\boldsymbol{v}) - \Gamma(\boldsymbol{v}^*)\|_\infty \\
&\geq \|\boldsymbol{v} - \Gamma(\boldsymbol{v})\|_\infty - \gamma \|\boldsymbol{v} - \boldsymbol{v}^*\|_\infty \\
\Rightarrow (1 + \gamma)\|\boldsymbol{v} - \boldsymbol{v}^*\|_\infty &\geq \|\Gamma(\boldsymbol{v}) - \boldsymbol{v}\|_\infty
\end{aligned}
$$

**Proof of Theorem 1** From lemma 1 we know there exists a policy $\pi$ and a $\tilde{\delta} > 0$ such that the optimal Bellman operator is a linear function on $U_{\tilde{\delta}}(\boldsymbol{v}^*)$. We now set $\delta$ sufficiently small such that

$$\frac{km(1+\gamma)}{1-\gamma}\|\boldsymbol{v}^{(0)} - \boldsymbol{v}^*\|_\infty < \frac{km(1+\gamma)}{1-\gamma}\delta < \tilde{\delta}.$$

The result is trivial for the first $k-1$ steps, which are performed exactly by standard value iteration. When $t > k$, we prove the result by induction. Suppose the conclusion is correct for previous steps, then we have

$$\|\sum_{i=1}^k \alpha_i^{(t)}\boldsymbol{v}^{(t-i)} - \boldsymbol{v}^*\|_\infty \leq \sum_{i=1}^k |\alpha_i^{(t)}|\|\boldsymbol{v}^{(t-i)} - \boldsymbol{v}^*\|_\infty \leq \sum_{i=1}^k |\alpha_i^{(t)}|\frac{1}{1-\gamma}\|B(\boldsymbol{v}^{(t-i)})\|_\infty$$

$$\leq \sum_{i=1}^k |\alpha_i^{(t)}|\frac{1}{1-\gamma}\|B(\boldsymbol{v}^{(0)})\|_\infty \leq \frac{km}{1-\gamma}\|B(\boldsymbol{v}^{(0)})\|_\infty$$

$$\leq \frac{km(1+\gamma)}{1-\gamma}\|\boldsymbol{v}^{(0)} - \boldsymbol{v}^*\|_\infty < \frac{km(1+\gamma)}{1-\gamma}\delta < \tilde{\delta}$$

It follows that

$$\|\boldsymbol{v}^{(t)} - \boldsymbol{v}^*\|_\infty = \|\Gamma(\sum_{i=1}^k \alpha_i^{(t)}\boldsymbol{v}^{t-i}) - \Gamma(\boldsymbol{v}^*)\|_\infty \leq \gamma\|\sum_{i=1}^k \alpha_i^{(t)}\boldsymbol{v}^{(t-i)} - \boldsymbol{v}^*\|_\infty$$

$$\leq \|\sum_{i=1}^k \alpha_i^{(t)}\boldsymbol{v}^{(t-i)} - \boldsymbol{v}^*\|_\infty < \tilde{\delta}$$

Therefore, $\sum_{i=1}^k \alpha_i^{(t)}\boldsymbol{v}^{(t-i)} \in U_{\tilde{\delta}}(\boldsymbol{v}^*)$, $\boldsymbol{v}^{(t)} \in U_{\tilde{\delta}}(\boldsymbol{v}^*)$, which implies

$$\Gamma(\sum_{i=1}^k \alpha_i^{(t)}\boldsymbol{v}^{(t-i)}) = \boldsymbol{r}_{\pi^*} + \gamma P_{\pi^*}\sum_{i=1}^k \alpha_i^{(t)}\boldsymbol{v}^{(t-i)}, \Gamma(\boldsymbol{v}^{(t)}) = \boldsymbol{r}_{\pi^*} + \gamma P_{\pi^*}\boldsymbol{v}^{(t)}.$$

Then we can get

$$B(\boldsymbol{v}^t) = \boldsymbol{r}_{\pi^*} + (\gamma P_{\pi^*} - I)\boldsymbol{v}^{(t)}$$

$$= \boldsymbol{r}_{\pi^*} + (\gamma P_{\pi^*} - I)(\boldsymbol{r}_{\pi^*} + \gamma P_{\pi^*}\sum_{i=1}^k \alpha_i^{(t)}\boldsymbol{v}^{(t-i)})$$

$$= \sum_{i=1}^k \alpha_i^{(t)}\left(\boldsymbol{r}_{\pi^*} + (\gamma P_{\pi^*} - I)(\boldsymbol{r}_{\pi^*} + \gamma P_{\pi^*}\boldsymbol{v}^{(t-i)})\right)$$

$$= \sum_{i=1}^k \alpha_i^{(t)}\gamma P_{\pi^*}\left(\boldsymbol{r}_{\pi^*} + \gamma P_{\pi^*}\boldsymbol{v}^{(t-i)} - \boldsymbol{v}^{(t-i)}\right)$$

$$= \gamma P_{\pi^*}\sum_{i=1}^k \alpha_i^{(t)}B(\boldsymbol{v}^{(t-i)})$$

Taking norm on both side of the equation and utilizing the definition of $\alpha_i^{(t)}, i = 1, 2, \cdots, k$, we get

$$\|\Gamma(\boldsymbol{v}^{(t)}) - \boldsymbol{v}^{(t)}\|_\infty \leq \gamma\|P_{\pi^*}\|_\infty\|\sum_{i=1}^k \alpha_i^{(t)} B(\boldsymbol{v}^{(t-i)})\|_\infty \leq \gamma\|B(\boldsymbol{v}^{(t-1)})\|_\infty = \gamma\|\Gamma(\boldsymbol{v}^{(t-1)}) - \boldsymbol{v}^{(t-1)}\|_\infty.$$

Therefore, we justify property (i). Property (ii) then follows directly from Lemma 2.

**Lemma 3.** *For any MDP, suppose the values $\boldsymbol{u}$ and $\boldsymbol{v}$ satisfy $\boldsymbol{u} \geq \boldsymbol{v}$, then*

$$\Gamma(\boldsymbol{u}) \geq \Gamma(\boldsymbol{v}).$$

**Proof**  As stated in section 2, $\boldsymbol{u} \geq \boldsymbol{v}$ means that $\boldsymbol{u}(s) \geq \boldsymbol{v}(s)$ for any state $s$. Suppose $\tilde{\pi} = \arg\max_\pi \boldsymbol{r}_\pi + \gamma P_\pi \boldsymbol{v}$, therefore for any $s$ we have that

$$\Gamma(\boldsymbol{u})(s) \geq \Gamma_{\tilde{\pi}}(\boldsymbol{u})(s) \geq \Gamma_{\tilde{\pi}}(\boldsymbol{v})(s) = \Gamma(v)(s).$$

**Proof of Theorem 2**  On the one hand,

$$\Gamma(\boldsymbol{v}^{(t)}) - \boldsymbol{v}^{(t)} = \max_\pi(\boldsymbol{r}_\pi + \gamma P_\pi \boldsymbol{v}^{(t)}) - \max_\pi(\boldsymbol{r}_\pi + \gamma P_\pi \sum_{i=1}^k \alpha_i^{(t)} \boldsymbol{v}^{(t-i)})$$

$$\leq \boldsymbol{r}_{\pi^{(t)}} + \gamma P_{\pi^{(t)}} \boldsymbol{v}^{(t)} - \boldsymbol{r}_{\pi^{(t)}} - \gamma P_{\pi^{(t)}} \sum_{i=1}^k \alpha_i^{(t)} \boldsymbol{v}^{(t-i)}$$

$$= \gamma P_{\pi^{(t)}}(\max_\pi(\boldsymbol{r}_\pi + \gamma P_\pi \sum_{i=1}^k \alpha_i^{(t)} \boldsymbol{v}^{(t-i)}) - \sum_{i=1}^k \alpha_i^{(t)} \boldsymbol{v}^{(t-i)})$$

$$\leq \gamma P_{\pi^{(t)}} \sum_{i=1}^k \alpha_i^{(t)}(\max_\pi(\boldsymbol{r}_\pi + \gamma P_\pi \boldsymbol{v}^{(t-i)}) - \boldsymbol{v}^{(t-i)})$$

$$= \gamma P_{\pi^{(t)}} \sum_{i=1}^k \alpha_i^{(t)} B(\boldsymbol{v}^{(t-i)})$$

On the other hand, we denote $\tilde{\pi} = \operatorname{argmax}_\pi(\boldsymbol{r}_\pi + \gamma P_\pi \sum_{i=1}^k \alpha_i^{(t)} \boldsymbol{v}^{(t-i)})$, then

$$\boldsymbol{v}^{(t)} - \Gamma(\boldsymbol{v}^{(t)}) = \max_\pi(\boldsymbol{r}_\pi + \gamma P_\pi \sum_{i=1}^k \alpha_i^{(t)} \boldsymbol{v}^{(t-i)}) - \max_\pi(\boldsymbol{r}_\pi + \gamma P_\pi \boldsymbol{v}^{(t)})$$

$$\leq \boldsymbol{r}_{\tilde{\pi}} + \gamma P_{\tilde{\pi}} \sum_{i=1}^{(k)} \alpha_i^{(t)} \boldsymbol{v}^{(t-i)} - \boldsymbol{r}_{\tilde{\pi}} - \gamma P_{\tilde{\pi}} \boldsymbol{v}^{(t)}$$

$$= \gamma P_{\tilde{\pi}}(\sum_{i=1}^k \alpha_i^{(t)} \boldsymbol{v}^{(t-i)} - \max_\pi(\boldsymbol{r}_\pi + \gamma P_\pi \sum_{i=1}^k \alpha_i^{(t)} \boldsymbol{v}^{(t-i)}))$$

$$= -\gamma P_{\tilde{\pi}} B(\sum_{i=1}^k \alpha_i^{(t)} \boldsymbol{v}^{(t-i)})$$

The above two results shows

$$\gamma P_{\tilde{\pi}} B(\sum_{i=1}^{k} \alpha_i^{(t)} \boldsymbol{v}^{(t-i)}) \leq B(\boldsymbol{v}^{(t)}) \leq \gamma P_{\pi^{(t)}} \sum_{i=1}^{k} \alpha_i^{(t)} B(\boldsymbol{v}^{(t-i)}).$$

Now, consider the rejection step. First we show that if $\boldsymbol{v} \in V_B$ then $\Gamma(\boldsymbol{v}) \in V_B$. If $\boldsymbol{v} \in V_B$, then $\Gamma(\boldsymbol{v}) \geq \boldsymbol{v}$. With Lemma 3, we have that $\Gamma(\Gamma(\boldsymbol{v})) \geq \Gamma(\boldsymbol{v})$, i.e. $\Gamma(\boldsymbol{v}) \in V_B$.

Next we show that if $\boldsymbol{v}^{(i)} \in V_B$ for $i < t$, then $\boldsymbol{v}^{(t)} \in V_B$. According to the rejection algorithm, we have $\boldsymbol{v}^{(t)} = \Gamma(\sum_{i=1}^{k} \alpha_i^{(t)} \boldsymbol{v}^{(t-i)})$. If $\Gamma(\sum_{i=1}^{k} \alpha_i^{(t)} \boldsymbol{v}^{(t-i)}) \geq \sum_{i=1}^{k} \alpha_i^{(t)} \boldsymbol{v}^{(t-i)}$, then $\sum_{i=1}^{k} \alpha_i^{(t)} \boldsymbol{v}^{(t-i)} \in V_B$ and $\boldsymbol{v}^{(t)} \in V_B$. If $\Gamma(\sum_{i=1}^{k} \alpha_i^{(t)} \boldsymbol{v}^{(t-i)}) < \sum_{i=1}^{k} \alpha_i^{(t)} \boldsymbol{v}^{(t-i)}$, then $\boldsymbol{v}^{(t)} = \Gamma(\boldsymbol{v}^{(t-1)})$ due to the rejection step. Since $\boldsymbol{v}^{(t-1)} \in V_B$, we have that $\boldsymbol{v}^{(t)} \in V_B$. Therefore, we also have that $B(\sum_{i=1}^{k} \alpha_i^{(k)} \boldsymbol{v}^{(t-i)}) \geq 0$. We have that

$$\|\Gamma(\boldsymbol{v}^{(t)}) - \boldsymbol{v}^t\| = \|B(\boldsymbol{v}^{(t)})\| \leq \gamma \|P_{\pi^{(t)}}\| \| \sum_{i=1}^{k} \alpha_i^{(t)} B(\boldsymbol{v}^{(t-i)})\|$$

$$\leq \gamma \|B(\boldsymbol{v}^{(t-1)})\| = \gamma \|\Gamma(\boldsymbol{v}^{(t-1)}) - \boldsymbol{v}^{(t-1)}\|$$

The second inequality is due to the definition of $\alpha_i^{(t)}$, $i = 1, 2, ..., k$.

**Proof of Theorem 3**   We prove the conclusion by induction. It is evident that the conclusion holds for the first $k - 1$ steps. Suppose the conclusion holds for the first $t - 1$ steps, then

$$\boldsymbol{v}^{(t)} = \max_{\pi}(\boldsymbol{r}_\pi + \gamma P_\pi \sum_{i=1}^{k} \alpha_i^{(t)} \boldsymbol{v}^{(t-i)})$$

$$\geq \boldsymbol{r}_{\tilde{\pi}} + \gamma P_{\tilde{\pi}}(\sum_{i=1}^{k} \alpha_i^{(t)} \boldsymbol{v}^{(t-i)})$$

$$\geq \boldsymbol{r}_{\tilde{\pi}} + \gamma P_{\tilde{\pi}} \boldsymbol{v}^{(t-1)}$$

$$\geq \boldsymbol{v}^{(t-1)},$$

where $\tilde{\pi} = \arg\max_{\pi} \boldsymbol{r}_\pi + \gamma P_\pi \boldsymbol{v}^{(t-1)}$. The second inequality comes from the extrapolation restriction. The Third inequality is due to that if $\boldsymbol{v} \in V_B$ then $\Gamma(\boldsymbol{v}) \in V_B$, which is shown in Theorem 2.

As shown in Theorem 2, we have that $\boldsymbol{v}^{(t)} \in V_B$ and $\sum_{i=1}^{k} \alpha_i^{(t)} \boldsymbol{v}^{(t-i)} \in V_B$. Therefore, $\boldsymbol{v}^{(t)} \leq \boldsymbol{v}^*$.

$$\boldsymbol{v}^* - \boldsymbol{v}^{(t)} = \boldsymbol{v}^* - \max_{\pi}(\boldsymbol{r}_\pi + \gamma P_\pi \sum_{i=1}^{k} \alpha_i^{(t)} \boldsymbol{v}^{(t-i)})$$

$$\leq \boldsymbol{v}^* - (\boldsymbol{r}_{\pi^*} + \gamma P_{\pi^*} \sum_{i=1}^{k} \alpha_i^{(t)} \boldsymbol{v}^{(t-i)})$$

$$= \gamma P_{\pi^*} \sum_{i=1}^{k} \alpha_i^{(t)} (\boldsymbol{v}^* - \boldsymbol{v}^{(t-i)})$$

$$\leq \gamma P_{\pi^*} (\boldsymbol{v}^* - \boldsymbol{v}^{(t-1)})$$

Taking the infinite norm on both sides, we get

$$\|\boldsymbol{v}^* - \boldsymbol{v}^{(t)}\|_\infty \leq \gamma\|P_{\pi^*}\|_\infty\|\boldsymbol{v}^* - \boldsymbol{v}^{(t-1)}\|_\infty \leq \gamma\|\boldsymbol{v}^* - \boldsymbol{v}^{(t-1)}\|_\infty,$$

which completes the proof.

## D. EXPERIMENT DETAILS

### D.1 MODEL ARCHITECTURE AND HYPER-PARAMETERS

For our experiments, we used the DQN(Mnih et al., 2013) architecture, where the Q-value network is composed of 3 convolutional layers, 1 fully connected layer, and 1 output fully connected layer. Each layer except the final layer is followed with a rectified linear activation(ReLU). The first convolutional layer use 32 $8 \times 8$ filters with stride 4, the second has 64 $4 \times 4$ filters with stride 2, and the third convolutional layer has 64 $3 \times 3$ filters with stride 1. The fully connected layer consists of 512 units and the final layer outputs a single value for each action. We used the Adam optimizer with learning rate 0.0001 and $\epsilon = 0.0015$. The discount was set to $\gamma = 0.99$. Training is done over 20M or 40M frames. We updated the target networks every 10000 steps. The size of experience replay buffer is 100000 tuples, where 32 mini batches were sampled every 4 steps to update the network. The exploration policy is $\varepsilon$-greedy policy with fixed $\varepsilon = 0.01$.

### D.2 PREPROCESSING OF ENVIRONMENTS

We preprocess the environment in the same way as the original DQN paper (Mnih et al., 2013) does. We utilize the action repeat technique, i.e., each action is repeated for the next four consecutive frames. The frames are firstly grey-scaled and then rescaled to the size of $84 \times 84$ pixels. Each state is represented by a concatenation of 4 consecutive frames. We fix all positive rewards to be 1 and all negative rewards to be -1, leaving 0 rewards unchanged. Transitions associated with the loss of a life are considered terminal.

## E. SOLVING (5) WHEN USING THE $\ell_2$ NORM AND TOTAL SPACE CONSTRAINT

Under the given setting, we may rewrite the original problem in the following form,

$$\text{minimize } \boldsymbol{\alpha}^\top (\hat{B}^{(t)})^\top \hat{B}^{(t)} \boldsymbol{\alpha}$$
$$\text{subject to } \mathbf{1}^\top \boldsymbol{\alpha} = 1.$$

This problem can be directly solved with an application of Lagrange multiplier method, namely, let $\lambda$ be the Lagrange multiplier, then we solve the problem

$$\max_{\boldsymbol{\alpha}} \boldsymbol{\alpha}^\top (\hat{B}^{(t)})^\top \hat{B}^{(t)} \boldsymbol{\alpha} + \lambda(\mathbf{1}^\top \boldsymbol{\alpha} - 1)$$

whose solution can be written explicitly as $\boldsymbol{\alpha} = -\frac{\lambda}{2}[(\hat{B}^{(t)})^\top \hat{B}^{(t)}]^{-1}\mathbf{1}$. Combine this result with the constraint, we can get $-\frac{\lambda}{2} = \frac{1}{\mathbf{1}^\top[(\hat{B}^{(t)})^\top \hat{B}^{(t)}]^{-1}\mathbf{1}}$, which implies $\boldsymbol{\alpha} = \frac{[(\hat{B}^{(t)})^\top \hat{B}^{(t)}]^{-1}\mathbf{1}}{\mathbf{1}^\top[(\hat{B}^{(t)})^\top \hat{B}^{(t)}]^{-1}\mathbf{1}}$.

