# OpenReview forum: "Accelerated Value Iteration via Anderson Mixing"
_ICLR.cc/2019/Conference_

### Official Review · AnonReviewer1 · 2018-11-01
**Accelerated Value Iteration via Anderson Mixing**

**Rating:** 4
**Confidence:** 3

**Review:**

This paper seems like a nice idea, but I'm not sure if it's ready for publication. It seems that the main contribution of this paper is the DA2Q algorithm, since the A2VI algorithm is a straightforward application of AA to VI. However the numerical examples are very weak, only 3 games are tested, and the results are not that strong. Furthermore in Figure 3 with the results it's not clear what the 'Time' axis is.

Smaller comments:

It seems like this recent paper should be cited:
https://arxiv.org/abs/1808.03971
it includes value iteration as an example, both in theory and in practice.

I think that lemma 1 is a direct consequence of the fact that PI has finite convergence (this is easily seen since there are finite policies and it converges).

In the contraction for PI what is K?

With the constraints as specified after equation 5 it is no longer Anderson acceleration. The convex combination constraint is just the standard alpha >= 0 constraint.

Rejection step seems very onerous, how often does it occur in practice?

Note that a simple application of AA to VI would not have the problem that it needs to "Jump out of the subspace".

DA2Q algorithm as printed is very complicated, can it be simplified somehow? Just focusing on the novel steps would help.

---

> ### Author Response · Authors · 2018-11-25
> **Response to AnonReviewer1**
>
> Thank you for your helpful comments.
>
> Q1: It seems that the main contribution of this paper is the DA2Q algorithm since the A2VI algorithm is a straightforward application of AA to VI.
>
> A1: Our approach is not a straightforward application of the Anderson Acceleration (AA) to VI. AA has the updating rule $v^t = \sum_i \alpha_i B(v^{t-i})$, while A2VI puts the weighted sum into the fixed-point iteration $v^t = B( \sum_i \alpha_i v^{t-i} )$, as stated below the Algorithm 1. Therefore, we get two advantages:
> (1) a geometric interpretation of the proposed approach;
> (2) a better convergence analysis than AA on the problem of value iteration.
>
> To see (2), note that Toth et al. [1] conducted the local convergence analysis for Anderson acceleration in nonlinear problems. They proved the Anderson iteration converges linearly with factor r. Applying their analysis to the problem of value iteration, the convergence rate r is larger than the discount $\gamma$ for $k\geq 2$ (worse than fixed-point iteration in the theoretical analysis). We get a better analysis with an exact convergence rate $\gamma$ (better than the theoretical result in [1]).
>
> [1] Toth, Alex, and C. T. Kelley. Convergence analysis for Anderson acceleration. SIAM Journal on Numerical Analysis 53.2 (2015): 805-819.
>
>
> Q2: In the contraction for PI what is K?
>
> A2: We give a clear definition to K in the revised version.
> $K$ is a constant to bound the ratio between the residual norm of probability transition matrix and the residual norm of state value vector. Please see Section (6.4.4) in the reference paper [2] for the detailed definition, i.e., there exists a $K$, $0 < K < \infty$, \|P_{v^t} - P_{v^*} \| \leq K \| v^t - v^* \|,where $\{v^t\}$ is the sequence of values generated by policy iteration, and $P_{v^t}$ is the transition probability induced by the state value $v^t$.
>
> [2] Puterman, Martin L. Markov decision processes: discrete stochastic dynamic programming. John Wiley \& Sons, 2014.
>
>
> Q3: Rejection step seems very onerous, how often does it occur in practice?
>
> A3: It occurs on a small frequency in practice. The numerical experiments show that without the rejection step the proposed algorithm still has a quite faster convergence rate than the original VI. In some environments, the rejection step can bring further improvement to convergence.
>
>
> The reviewer presented a very recent paper on arxiv (https://arxiv.org/abs/1808.03971) related to the proposed approach. We list some major differences listed here:
> (1) That paper studies the contraction function on real values, while we care about the state values on discrete state space. In the discrete setting, the $\max$ operation make the state values not have the non-expansive property. Therefore, the convergence analysis of that paper could not be applied to our cases.
>
> (2) That paper focuses on the convergence analysis on the type-I Anderson Acceleration, while our approach is more related to the type-II Anderson Acceleration. Moreover, our approach is not a straightforward application of AA to VI. AA has the updating rule $v^t = \sum_i \alpha_i B(v^{t-i})$, while A2VI puts the weighted sum into the fixed-point iteration $v^t = B( \sum_i \alpha_i v^{t-i})$, as stated below the Algorithm 1. In this way, we get two advantages: (a) the geometric interpretation of the algorithm; (b) making the convergence analysis tractable.
>
> (3) Consider the theoretical analysis. That paper studied the type-I AA for the general non-smooth optimization problem. They provide a convergence analysis as the iteration step $n$ goes to infinity, but without convergence rate analysis. We give global linear convergence rate analysis with the rejection step in Theorem 2 and Theorem 3, locally linear convergence rate analysis without the rejection step in Theorem 1.

---

> > ### Public Comment · (anonymous) · 2018-12-06
> > **quick comment on the paper https://arxiv.org/abs/1808.03971**
> >
> > Just one quick comment on the paper https://arxiv.org/abs/1808.03971 (Zhang, O’Donoghue and Boyd paper (2018)). After reading it I think the example about value iteration in their paper is taking the value function as the iteration variable, which is a continuous real-valued vector of length |S|. They are also talking about discrete state and action space. The non-expansive (actually gamma-contractive) property of the bellman operator is thus the same as the one discussed in your paper. Please correct me if I misunderstood. Thanks!
> >
> > The approach is of course not the same at all, and in particular it is about general fixed point problems and type-I AA, and it is not about rate analysis. But the safe-guarding idea seems to be related to the rejection step here, and in particular it is related to accelerating VI via AA. It might also be interesting to see how it performs if one replaces the type-II AA in your paper with type-I AA in VI and DQN. I would suggest adding at least a short conceptual comparison with the paper https://arxiv.org/abs/1808.03971 in your draft and mention the potential consideration of applying type-I AA for accelerated RL to shed light on future work.

---

> > > ### Author Response · Authors · 2018-12-09
> > > **Response**
> > >
> > > Yes, you are correct. Thank you for your helpful suggestions. And we have added a comparison with the paper \url{https://arxiv.org/abs/1808.03971}.
> > >
> > > Thank you for your agreement that the above paper does not discuss convergence rate analysis while we conduct rate analysis in local and global cases for A2VI.
> > > Although it seems that the safe-guarding idea is related to our rejection step, they are quite different.
> > > The safe-guarding idea depends on two theoretical constants $D$ and $\epsilon$, which are difficult to set appropriate values.
> > > The rejection step in our method has a straightforward way to guarantee the monotonicity.
> > > Further, our approach is not a straightforward application of type-II AA to VI.
> > > The updating rule in type-II AA to VI is $v^t = \sum_i \alpha_i B(v^{t-i})$.
> > > We put the weighted sum into the fixed-point iteration and get the updating rule $v^t = B( \sum_i \alpha_i v^{t-i})$.
> > > In this way, we get two advantages: (1) the geometric interpretation of the algorithm; (2) making the convergence analysis tractable.
> > >
> > > BTW, the original version of this work is finished in May 2018, but we did not make it available. Thanks!

---

> > > > ### Public Comment · (anonymous) · 2018-12-14
> > > > **Agreed**
> > > >
> > > > Thanks a lot for the response and further explanations. Yes I completely agree with your comments on the difference, and in particular, the difference between the safeguards in that paper and the one in your paper.
> > > >
> > > > Btw, in the paper of Zhang et al., the safe-guards are enforced on the residuals for the general fixed-point problems does not seem to be of central usefulness in practice, as they simply set D to be 1e6 and \epsilon to be 1e-6. The other two tricks that ensures uniform bounds on the approximate Jacobians seem to be more central both in theory and practice. Have you observed the rejection step in your paper to be practically important?

---

### Official Review · AnonReviewer3 · 2018-11-02
**Extension to the Approximate DP case needed**

**Rating:** 4
**Confidence:** 4

**Review:**

This paper introduces the "Anderson mixing" ideas from the broader literature on general fixed-point problems to the specific problem of finding the fixed-point to the Bellman optimality equations for a Markov Decision Processes. The general idea is to summarizes the history of previous iterates (value functions in this case) by finding of convex combination which also minimizes the residuals. The authors provide a solution for when an iterate is no longer representable by a convex combination of the recent history by simply bypassing the interpolation step and replacing it with a usual value iteration step. Using the intuition developed in the MDP case, they then adapt their DP algorithms to the learning case by substituting exact (tabular) value functions with deep function approximators. Experimental results are presented in 3 games from the ALE environment.

The jump from the DP formulation to the learning case is rather abrupt, and lacks sufficient motivation. The way the paper is currently structured is 50-50: 50% of the contribution is the DP view of the proposed method while the remaining half comes from the deep formulation (and experiments). I think that I would have preferred to see the entire paper being dedicated to the DP point of view, followed by a more principled Approximate DP analysis in the simpler linear case. Dedicating the remaining of the paper to the deep formulation almost feels like a missed opportunity to fully developing the theory initiated in the first section. But then of course the price to pay would be a paper which would be less aligned with the "representation learning" aspect of the conference. My main concern is that extending this technique to the deep setting mare involve some serious interference with other mechanisms already at play. It is very difficult to explain if the observed improvement come from the underlying DP basis or as a secondary effect of architectural and algorithmic considerations.

To my knowledge, this is the first attempt at using Anderson mixing in the MDP framework. However, I would appreciate if the authors could survey previous attempts (if any) by other authors, or more generally existing results in the literature on non-linear fixed-point methods.  You may find relevant work by consulting the recent Zhang, O’Donoghue and Boyd paper (2018).

# Detailed comments

> Puterman 2014

The 2014 edition is likely to be a re-print of the 1994 which is commonly cited. I would double-check to see if there is any difference in the content between the 2014 and 1994 edition. If not (and just a re-print) I would cite the 1994 edition which is more widely recognized.

> Citations for VI and PI
You should cite Bellman 1957 and Howard 1961 (not Puterman). For exact references, see bibliographical remarks in Puterman.

> Citation for Modified policy iteration

Please cite original paper(s) by Puterman and Brumelle ~1978. See bibliographical remarks in Puterman 1994 (or 2014) for the origins of MPI.

>  via the Neumann expansion

truncated

> computationally inefficient for complex decision problems

Compared to what? More efficient than full PI for sure

> Page 2, notation for $\Gamma_\pi$ vs $\Gamma$

I suggest using a different notation for the (linear) policy evaluation operator vs the Bellman optimality one. The subscript "_\pi$ is easy to miss.

> converges much faster with K

Define K

> In most cases, we can

In reinforcement learning, we can

> value iteration can be finished

Finished ?

> value iteration can be finished by estimating Γ(v) through sampling.

We are no longer in the realm of DP, but more stochastic approximation methods. This isn't quite VI anymore. I would be more careful when jumping from one setting to the other.

> provided the sampling estimations are accurate enough

The approach described so far does not involve any sampling.

> This modification is based on the observation that the recent successive policies do not

So far, the mixing equations (3) and (4) only describe the evaluation case. You haven't mentioned yet how you plan to combine this into a more general control algorithm where successive (changing) policies are generated.

> the solution can be written explicitly as

Please cite where this comes from (or provide proof inline or appendix)

> while PI is similar to Newton’s method

Cite Puterman and Brumelle for the original work on showing the connection between PI and Newton's method.

> except that the tangent line is replaced with a secant line.

Please explain this intuition: how you obtain this geometric interpretation.
Also, the secant method being an analogue to quasi-Newton methods, and policy iteration being Newton's method, there is an opportunity to better develop and explain those parallels.

---

> ### Author Response · Authors · 2018-11-25
> **Response to AnonReviewer3**
>
> Thanks for your helpful comments.
>
> Q1: Extending this technique to the deep setting may involve some serious interference with other mechanisms. It is difficult to explain if the observed improvement comes from the underlying DP basis.
>
> A1: DA2Q and DQN have a quite similar algorithm flow as stated in Algorithm 2. We only add the Anderson mixing to the target value estimation and fix the other parts of the algorithm flow. The comparison experiments are conducted on the same environment settings and the same parameters. The numerical experiments have shown a significant improvement of the proposed approach over the original VI.
>
> Q2: Detailed writing revision.
>
> A2: Thank you very much for your helpful comments on the citation and some statements. We carefully read your comments and revise the original version.
>
>
> The reviewer presented a very recent paper on arxiv (https://arxiv.org/abs/1808.03971) related to the proposed approach. We list some major differences listed here:
> (1) That paper studies the contraction function on real values, while we care about the state values on discrete state space. In the discrete setting, the $\max$ operation make the state values not have the non-expansive property. Therefore, the convergence analysis of that paper could not be applied to our cases.
>
> (2) That paper focuses on the convergence analysis on the type-I Anderson Acceleration, while our approach is more related to the type-II Anderson Acceleration. Moreover, our approach is not a straightforward application of AA to VI. AA has the updating rule $v^t = \sum_i \alpha_i B(v^{t-i})$, while A2VI puts the weighted sum into the fixed-point iteration $v^t = B( \sum_i \alpha_i v^{t-i})$, as stated below the Algorithm 1. In this way, we get two advantages: (a) the geometric interpretation of the algorithm; (b) making the convergence analysis tractable.
>
> (3) Consider the theoretical analysis. That paper studied the type-I AA for the general non-smooth optimization problem. They provide a convergence analysis as the iteration step $n$ goes to infinity, but without convergence rate analysis. We give global linear convergence rate analysis with the rejection step in Theorem 2 and Theorem 3, locally linear convergence rate analysis without the rejection step in Theorem 1.

---

### Official Review · AnonReviewer2 · 2018-11-05

**Rating:** 7
**Confidence:** 4

**Review:**

This is a very well-written paper which proposed a way to accelerate the value-iteration of MDP. The method is the so-called "Anderson-Mixing" method. It replaces the policy evaluation step by solving a smaller linear equation: find a linear combination of a few historical values to represent the value of the current policy. The paper also presents a very nice explanation of why such a modification of VI accelerates VI. The paper also extends the method to DQN and shows a very nice acceleration. The experiments are convincing and interesting.

I only have two concerns:

1) In section 4, the convergence proof is shown but the contraction is only gamma. This is the same as the original VI. Of course, this is the worst case best bound. Is it possible to show a result that the modified-VI is always better than the original VI?

2) In section 4, the dependence on k has not been studied. But k actually critically affects the time complexity. Is it possible to obtain convergence proof depending on k?

---

> ### Author Response · Authors · 2018-11-25
> **Response to AnonReviewer2**
>
> Thanks for your helpful comments!
>
> Q1: Is it possible to obtain convergence proof depending on k? Is it possible to show a result that the modified-VI is always better than the original VI?
>
> A1: As we know, no analysis of AA implies that larger k has better convergence result rigorously although the empirical results support this point.
> In the numerical experiments, the modified-VI is always better than VI, but it is difficult to theoretically analyze whether the modified VI is always better than the original VI.
>
> Toth et al. [1] conducted the local convergence analysis for Anderson acceleration in nonlinear problems.
> They proved the Anderson iteration converges linearly with factor r.
> Applying their analysis to the problem of value iteration, the convergence rate r is larger than the discount $\gamma$ for $k \geq 2$ (worse than fixed-point iteration in the theoretical analysis).
> We get a better analysis than the linear convergence rate is exact $\gamma$ (better than the theoretical result in [1]).
>
> [1] Toth, Alex, and C. T. Kelley. Convergence analysis for Anderson acceleration. SIAM Journal on Numerical Analysis 53.2 (2015): 805-819.

---

### Public Comment · ~Matthieu_Geist1 · 2018-11-08
**Another paper on Anderson acceleration for RL**

This ICLR submission is a very interesting paper.

In addition to the reference provided by two of the reviewers, the authors might be interested by this (very recent) workshop paper, that also proposes to use Anderson acceleration for RL : https://arxiv.org/abs/1809.09501

The paper is much more preliminary than this ICLR submission (eg, no convergence analysis, extension to deep reinforcement learning only briefly outlined), but it provides some complementary things (notably, a partial and empirical discussion of the actual speed of convergence of the sequence of greedy policies for accelerated VI).

---

> ### Author Response · Authors · 2018-11-25
> **Thanks for your helpful comments!**
>
>  Thanks for your sharing. We revise our paper and cite your work.

---

### Meta-Review · Area_Chair1 · 2018-12-14
**Interesting idea, but contributions need to be strengthened**

**Confidence:** 4
**Recommendation:** Reject

**Metareview:**

The paper proposes to use Anderson Mixing to accelerate value iteration and DQN.  The idea is interesting, with some theoretical and empirical support.  However, reviewers feel that the contribution is somewhat limited, and certain parts (e.g., the DP view) can be further developed to strengthen the technical contribution.  Furthermore, one reviewer points out that the empirical results are not very strong, where the improvements on 3 Atari games are not very substantial.  Overall, while the paper is interesting and does have the potential, it seems too preliminary to be published in its current form.

Minor comments:
1. The paper is partially motivated by the claim given at the beginning of section 3: "Based on the observation that full policy evaluation accelerates convergence, ..."  Can a reference be given?

2. Another way to look at Anderson Mixing is the standard linear value function approximation framework, where the previous K value functions serve as basis functions.  See Mahadevan & Maggioni (JMLR'07), Parr et al. (ICML'08) and Konidaris et al. (AAAI'11) for a few examples of constructing basis functions; the approach here seems to provide another way to automatically construct basic functions.  A discussion would be helpful.